# Effect of Interactions between Phosphorus and Light Intensity on Metabolite Compositions in Tea Cultivar Longjing43

**DOI:** 10.3390/ijms232315194

**Published:** 2022-12-02

**Authors:** Santosh KC, Lizhi Long, Qunfeng Zhang, Kang Ni, Lifeng Ma, Jianyun Ruan

**Affiliations:** 1Key Laboratory of Tea Plant Biology and Resources Utilization (Ministry of Agriculture), Tea Research Institute, Chinese Academy of Agricultural Sciences, Hangzhou 310008, China; 2Xihu National Agricultural Experimental Station for Soil Quality, Hangzhou 310008, China

**Keywords:** *Camellia sinensis*, phosphorus, light intensity, metabolome, green tea

## Abstract

Light intensity influences energy production by increasing photosynthetic carbon, while phosphorus plays an important role in forming the complex nucleic acid structure for the regulation of protein synthesis. These two factors contribute to gene expression, metabolism, and plant growth regulation. In particular, shading is an effective agronomic practice and is widely used to improve the quality of green tea. Genotypic differences between tea cultivars have been observed as a metabolic response to phosphorus deficiency. However, little is known about how the phosphorus supply mediates the effect of shading on metabolites and how plant cultivar gene expression affects green tea quality. We elucidated the responses of the green tea cultivar Longjing43 under three light intensity levels and two levels of phosphorus supply based on a metabolomic analysis by GC×GC-TOF/MS (Two-dimensional Gas Chromatography coupled to Time-of-Flight Mass Spectrometry) and UPLC-Q-TOF/MS (Ultra-Performance Liquid Chromatography-Quadrupole-Time of Flight Mass Spectrometry), a targeted analysis by HPLC (High Performance Liquid Chromatography), and a gene expression analysis by qRT-PCR. In young shoots, the phosphorus concentration increased in line with the phosphate supply, and elevated light intensities were positively correlated with catechins, especially with epigallocatechin of Longjing43. Moreover, when the phosphorus concentration was sufficient, total amino acids in young shoots were enhanced by moderate shading which did not occur under phosphorus deprivation. By metabolomic analysis, phenylalanine, tyrosine, and tryptophan biosynthesis (PTT) were enriched due to light and phosphorus effects. Under shaded conditions, *SPX2* (Pi transport, stress, sensing, and signaling), *SWEET3* (bidirectional sugar transporter), *AAP* (amino acid permeases), and *GSTb* (glutathione S-transferase b) shared the same analogous correlations with primary and secondary metabolite pathways. Taken together, phosphorus status is a crucial factor when shading is applied to increase green tea quality.

## 1. Introduction

Tea (*camellia sinensis* L.), of the Theaceae family, originates from China and has been used as a regular beverage for thousands of years. The tea plant is very adaptable to agroforestry [1]. These shade-tolerant plants possess beneficial bioactive tea components [2] that are classified as primary and secondary metabolites and determine the product quality [3]. Carbohydrates, amino acids, and flavonoids are among the highest quality components [4,5]. Thousands of metabolites present in each class are differently expressed due to various environmental and mineral effects [6]. The effects of light and phosphates, in particular, need to be studied to provide insights into how essential plant physiological activities vary with the growth and quality of young tea shoots.

It is well known that a phosphorus deficiency negatively impacts plant growth [7], increases root length [8], enhances secreted or root-associated acid phosphatase activities [9], reduces phosphorylated metabolite synthesis [10], and decreases leaf evapotranspiration [11]. Having an adequate light intensity sustains normal root development [12], and the minimum light amount cannot be lowered by increasing the concentration of nutrients [13]. An elevated light intensity influences the biosynthesis of secondary metabolites [14], while a low light intensity leads to the accumulation of flavonoids [15].

P is one of the driving elements in plant stoichiometric relations on the carbon-nutrient balance [16,17], and secondary metabolites might increase or decrease under low light intensity conditions in some plants [18]. Tea plant leaves cultivated in the shade contain high concentrations of amino acids due to sugar supply regulation and a variety of regulatory signals [2]. Light and nutrients are the major ecological and environmental factors that influence plant growth, biomass allocation, and other physiological responses. Insufficient light intensity leads to a decreased P uptake, and the interaction between light and nutrients significantly affects the biomass [19] and nutrient allocation. However, the relative growth rate differs amongst species [20].

Numerous genes have significant responses when plants are kept in the dark [14]. Light regulates photosynthetic carbon assimilation [21], whereas shading preserves amino acids from protein synthesis [16], provides energy for plants during periods of extended darkness [22], and supplies photosynthates that regulate gene expression [23]. Several genes affect carbohydrates in terms of stress responses, maintaining homeostasis during cellular signal transduction [6] and regulating carbohydrate anabolism and catabolism [24]. Glucose utilization runs parallel to glycolysis to reduce carbohydrate metabolites from undergoing further insufficient ammonium assimilation in the roots, as this results in an insufficient supply of carbohydrates [25,26]. This whole process depends upon the white light intensity, which regulates polyamines, inhibits flavanol biosynthesis through polymerization, and competes for substrate recruitment [27,28].

Furthermore, amino acids are responsible for tea quality. For instance, low-quality grades of black tea contain lower free amino acid contents [29]. The most abundant amino acids in tea are theanine (Thea), glutamine (Gln), glutamic acid (Glu), and arginine (Arg) [30,31]. The coefficient value obtained in one study revealed that Thea, Gln, and Arg are the major contributors to high-ranking teas, whereas aspartic acid (Asp) and alanine (Ala) are major contributors to low-ranking teas [4]. Plants regulate the carbon flux toward aromatic amino acid biosynthesis at the transcriptional and post transcriptional levels [5]. Plants retain their production of the biosynthesis of downstream products, which are often drastically changed in response to specific nutritional and environmental conditions.

Biosynthesis processes in tea plant organs respond differentially to variable light intensities and changes in the phosphorus supply, causing changes in specific metabolites. Gene expression for enzymes and transcriptional factors drives metabolites synthesis, but metabolites could also regulate gene expression and functionality by a feedback loop. Thus, the molecular mechanisms involved in primary and secondary metabolites due to plant nutrition and physiological responses at different irradiance levels have still not been clarified. Untargeted metabolomics profiling based on GC×GC-TOF/MS and UPLC-Q-TOF/MS integrated with targeted amino acid and catechins by HPLC was used to investigate the effects of different light and P levels on metabolites in mature leaves, roots, and young shoots of tea plants. To gain a deep understanding of the molecular mechanisms associated with specific pathways, transcription factors and genes for biosynthesis were monitored and collaboratively analyzed with metabolites. The study objective was to gain systematic information on the responses of pathways, their specific metabolites, and gene expression to light intensity and phosphorus and the effects of their interactions on the Longjing43 tea cultivar.

## 2. Results

### 2.1. Plant Biomass and Elemental Concentrations

The dry weights of young shoots, leaves and roots declined significantly due to P deficiency in full light (FL) and in 50% (ML) and 20% (LL) of full light conditions (Table 1). Shading treatment decreased the leaf biomass but increased the root biomass in ML compared with FL under both P-sufficient and P-deficient conditions. Thus, no significant difference was observed between ML and FL in terms of the total biomass. However, further shading decreased the total biomass in LL compared with ML and FL under both P conditions. The light regime and P level interactions were observed for young shoots, leaves, and root biomass. Young shoot and leaf P concentrations decreased under conditions of P deficiency, regardless of the light intensity treatment. With a decrease in the light intensity from FL to LL, the young shoot P concentration declined under both P-sufficient and P-deficient conditions (Table 1 and Figure 1).

In addition to the tissue P concentration, shading treatments also affected the concentrations of other elements. In young shoots, shading decreased the K, S, Ca, Al, Fe, B, Cu, and Mn concentrations but increased the Zn concentration under both P-sufficient and P-deficient conditions (Appendix A). In young shoots, LL treatment enhanced the Mg concentration compared with FL and ML when P was sufficient. In leaves, mild shading (ML) and severe shading (LL) had different effects on the Ca, Al, and Fe concentrations compared with FL under both P conditions (Appendix A). Shading decreased foliar B, Cu, and Zn concentrations but increased the Mn concentration with a greater influence under ML than LL. In both young shoots and leaves, significant interactions between light regimes and P levels were observed for the K, Ca, Mg, Fe, Cu, and Zn concentrations.

### 2.2. Comparative Effects of Untargeted Metabolites under Different Light Intensities and Phosphorus Levels

#### 2.2.1. Young Shoots

In the young shoots of Longjing43, primary metabolites were responsive to changes in the light intensity and P supply (Figure 2 and Appendix A). In inositol phosphate metabolism (IPM), 1d myo-inositol-3P, d-glucurontate and 1d myo-inositol-1P decreased in LL compared with FL under a P supply. P limitation decreased myo-inositol under all light conditions. In the pentose phosphate pathway (PPP), d-ribulose 5-phosphate (from 1.42 to 1.73; *p* < 0.001), d-ribose 5-phosphate (from 1.32 to 1.64; *p* < 0.001), and d-gluconate (from 1.36 to 1.71; *p* < 0.001) increased in ML compared with other light treatment groups, regardless of the P supply. Similarly, P deficiency decreased d-ribulose 5-phosphate and d-gluconate under all light treatments. In pentose and glucuronate interconversions (PGI), l-arabinose (from 1.57 to 1.84; *p* < 0.001) increased in ML compared with the FL and LL treatment groups. P deficiency increased the d-xylose concentration (from 1.06 to 1.56; *p* < 0.001) but decreased the l-arabinose concentration (from 1.57 to 0.63; *p* < 0.001) compared with the P supply under all light treatments. Regarding fructose and mannose metabolism (FMM), d-fructose (from 1.42 to 1.63; *p* < 0.001) and d-mannose (from 2.05 to 2.44; *p* < 0.001) increased in ML compared with other light treatment groups, regardless of the P status. P deficiency decreased the d-fructose concentration (*p* < 0.001) but increased the d-mannose 6-phosphate concentration in all light treatment groups. Regarding glycine and serine metabolism (GSM), glycine (from 1.51 to 1.81; *p* < 0.001) and l-homoserine (from 1.49 to 1.79; *p* < 0.001) increased in ML compared with the other light treatment groups. P deficiency decreased the l-serine (*p* < 0.001), glycine (*p* < 0.001), threonine (*p* < 0.001), and l-homoserine (*p* < 0.001) concentrations in the ML and LL treatment groups. In terms of phenylalanine, tyrosine, and tryptophan biosynthesis (PTT), l-tryptophan (from 1.61 to 2.49; *p* < 0.001), phenylalanine (from 1.6 to 1.85; *p* < 0.05), and l-tyrosine (from 1.71 to 1.48; *p* < 0.001) increased in ML compared with the other light treatment groups. P deficiency increased the l-tryptophan (*p* < 0.001) and phenylalanine concentrations (*p* < 0.001) in the ML and LL treatment groups but significantly reduced the l-tyrosine concentration in the ML (from 1.71 to 1.48; *p* < 0.001) and LL (from 1.16 to 0.74; *p* < 0.001) treatment groups. In terms of leucine, isoleucine, and valine biosynthesis (LIV), the l-leucine (from 1.5 to 1.74; *p* < 0.001), l-isoleucine (from 1.27 to 2.28; *p* < 0.001), and l-valine (from 1.26 to 1.65; *p* < 0.001) concentrations increased in the ML group compared with the other light treatment groups. P deficiency significantly increased the l-isoleucine concentration (from 1.47 to 2.28; *p* < 0.001) but decreased the l-leucine concentration (from 1.5 to 0.71; *p* < 0.001) in the ML and LL treatment groups and decreased the l-valine concentration in all light treatment groups. In the Kreb’s (TCA) cycle, the oxalic acid (from 1.45 to 2.29; *p* < 0.001), citrate (from 1.5 to 2.35; *p* < 0.001), isocitrate (from 1.5 to 2.32; *p* < 0.001), succinate (from 1.47 to 2.27; *p* < 0.001), fumarate (from 1.33 to 2.05; *p* < 0.001), and malate (from 1.33 to 2.05; *p* < 0.001) concentrations significantly increased in the ML compared with FL and LL groups but declined with −P treatment in the ML and LL groups. P deficiency increased the oxoglutarate concentration (from 1.06 to 2.46; *p* < 0.001) more than all light treatments. Surprisingly, in the FL treatment group, there was an insignificant effect on oxalic acid compared with the other TCA cycle metabolites. Similarly, in terms of alanine, asparate, and glutamate metabolism (AAG), the concentrations of l-asparate, l-asparagine, and l-glutamine decreased with both P deficiency and further shading (from ML to LL), but the l-alanine and l-glutamate concentrations increased due to P deficiency in the ML or LL light treatment groups. In terms of arginine and proline biosynthesis (APB), the l-proline concentration increased with P deficiency in the ML and LL groups, but the concentrations of hydroxyproline, l-arginine, ornithine, carbamoyl-P, and n-acetyl-l-citrulline (and citrulline) concentrations declined due to P deficiency in the ML or LL light treatment groups.

Generally, in penylpropanoid biosynthesis, shading decreased the caffeoyl shikimic acid concentration and increased the caffeoyl-CoA concentration, regardless of the P supply (Figure 3 and Appendix A). In the biosynthesis of flavonol glycosides, shading increased the quercetin concentration and decreased the isoquercetin concentration. P deficiency increased the isoquercetin (*p* < 0.001) and quercetin (*p* < 0.001) concentrations under all light treatments. In anthocyanidin and anthocyanin biosynthesis (AAB), shading significantly decreased the cyanidin (*p* < 0.001), leucocyanidin (*p* < 0.001), (+)-catechin (*p* < 0.001) and delphinidin (*p* < 0.001), procyanidin B1 (*p* < 0.001), and (−)-epigallocatechin (*p* < 0.001) concentrations compared with FL. P deficiency significantly increased the cyanidin, (+)-catechin and delphinidin concentrations but decreased the procyandin B1 and (−)-epigallocatechin concentrations in all light treatment groups. In flavone biosynthesis, shading decreased the kaempferol, malvidin, luteolin, and apigenin concentrations (*p* < 0.001) but increased the luteolin 7-*O* glucoside concentration (*p* < 0.001) compared with FL under a P supply. P deficiency increased the malvidin concentration (0.17 to 1.87; *p* < 0.001) but decreased the apigenin concentration (0.6 to 0.01; *p* < 0.001) in all light treatment groups.

#### 2.2.2. Leaves

In leaves, the myo-inositol, 1D-myo-inositol-3P, and 1D-myo-inositol-1P concentrations in IPM increased due to P deficiency under FL and ML conditions, while the D-Glucuronate concentration increased due to −P treatment under all light conditions (Appendix A). In terms of FMM, the d-mannose concentration increased under P deficient conditions under all light treatments, while the D-Fructose and D-Mannose-6P concentrations increased due to −P under FL conditions. In terms of PPP, the d-ribose5-phosphate (from 2.13 to 8.78; *p* < 0.05) and ribose (from 2.05 to 4.11; *p* < 0.05) concentrations increased under −P treatment conditions at all light intensities but decreased under ML compared to FL, regardless of the P supply. In terms of PGI, moderate shading decreased the l-arabitol, L-Arabinose, and L-Xylulose concentrations, while it increased the D-Xylulose-5P concentration compared to FL. In terms of GSM, l-homoserine (from 1.97 to 2.71; *p* < 0.001) increased under P deficient conditions under all light treatments. In terms of LIV, the l-leucine and l-valine concentrations increased due to P deficiency at all light intensities. In terms of PTT biosynthesis, moderate shading decreased the l-phenylalanine and 4-Hydroxy-phenylpyruvate concentrations compared to FL with a P supply. Regarding the Kreb’s cycle, moderate shading decreased the Oxalic acid, Citrate, and Malate concentrations compared with FL under a P supply. The fumarate concentration increased under ML but declined under LL compared to FL, regardless of the P supply. In terms of AAG, the l-asparagine and l-glutamine concentrations decreased under P deficient conditions under all light treatments. In terms of APB, the l-proline, carbonyl-1P, and n-acetyl-l-citrulline concentrations increased but the citrulline concentrations decreased under P deficient conditions under all light treatments.

Generally, in terms of phenylpropanoid biosynthesis, *p*-Coumaroyl quinic acid (from 21.71 to 58.46; *p* < 0.01) increased under P deficient conditions under all light treatments. In terms of anthocyanidin and anthocyanin biosynthesis, P deficiency increased the cyanidin (from 1 to 4.68; *p* < 0.001) and cyanidin 3-rutinoside (from 0.58 to 6.54; *p* < 0.001) concentrations, while it decreased the myricetin (from 1.87 to 0.03; *p* < 0.001), delphinidin (from 1.9 to 0.01; *p* < 0.01, (+)-gallocatechin (from 2.08 to 0.19; *p* < 0.001), and (−)-epigallocatechin (from 1.4 to 0.12; *p* < 0.001) concentrations. In flavone biosynthesis, P deficiency decreased the malvidin (from 0.47 to 0.67; *p* < 0.001) and luteolin 7-*O* glucoside (from 0.6 to 0.06; *p* < 0.01) concentrations under all light treatments.

### 2.3. Overview of Targeted Metabolomics Analysis of Tea Plants under P-Insufficient and P-Sufficient Conditions

#### 2.3.1. Young Shoots

In young shoots, the contents of most free amino acids and flavonoids were significantly altered by light and P treatments (Figure 4). Under the ML intensity, the Ser, Ile, Arg and Thea concentrations increased under +P conditions. However, shade decreased the Ser, Arg, and Tyr concentrations under −P conditions. P deficiency increased the concentrations of the most abundant amino acids, especially Thea (1.57, −P/+P), under all light levels (Figure 4), but shading decreased the Gly (0.58), His (0.67), and Met (0.20) concentrations. The Val (1.93), Phe (1.52), Asp (1.14), Pro (1.38), Cys (1.21), Leu (1.58), and Lys (1.05) concentrations increased due to shading, while P deficiency increased the Gly (1.94), Asp (1.47), Cys (1.74), His (2.25), and Lys (1.43) concentrations but decreased the Val (0.32), Phe (0.55), Pro (0.74), Leu (0.35), and Met (0.41) concentrations (Appendix A). The EGC concentrations of young shoots declined with shading (0.35) and −P (0.71) treatment. The catechin concentrations decreased due to shading (0.57) and −P (0.91) treatment (Appendix A). PCA loading treatments were directly linked to the correlation analysis for group separation (Figure 5 and Figure 6A). In young shoots, P nearly aligned with C and EC, and the biomass with EGC had high principal component contributions (Figure 5 and Figure 6A).

#### 2.3.2. Leaves

In leaves, shading increased the concentrations of the amino acids Arg (1.1), Glu (1.17), His (1.02), Lys (1.16), Pro (1.98), Ser (1.39), and Tyr (1.06) but decreased the concentrations of Asp (0.9), Cys (0.77), Ile (0.39), Met (0.62), Phe (0.41), Thr (0.32), and Val (0.53) (Appendix A). P deficiency increased the concentrations of the amino acids Arg (1.74), Asp (1.15), Glu (1.59), His (2.59), Lys (1.26), Ser (1.12), Thr (3.04), and Tyr (1.76) but decreased the concentrations of Cys (0.22), Ile (0.43), Met (0.46), Phe (0.33), Pro (0.28), and Val (0.22). Similarly, shading greatly increased the EGC (23.89) but decreased the EGCG (0.57). In leaves, P deficiency decreased the EGC (0.71) and decreased the EGCG (1.19). P was grouped with Ile and ECG and had a high principal component contribution (Appendix A). Biomass was almost aligned with EC and EGC but had a lower principal component contribution.

### 2.4. Relative Gene Expression in Response to P Limitation and Light Changes

The transcription factors *PHR1*, *PHO1*, and *SPX2* were upregulated under P deficient conditions in each light treatment group (Appendix A). The *PHO1* expression level was higher in LL than in FL and ML, while *SPX2* showed the highest expression level in ML under P deficient conditions. Highly significant effects of light, P level, and the interaction between the two were observed for *PHO1* and *SPX2*. Increased expression levels of *SWEET3* and *GSTb* were induced by P deficiency and shading. In contrast, *AAP* expression was downregulated by shading treatment. The expression levels of most primary-metabolite-related genes were upregulated by −P under all light treatments (Appendix A). Shading increased the expression of *IMPL1* under P-deficient conditions (Figure 6). Similarly, the expression levels of the transporter genes *SWEET3*, *AAP*, and *GSTb* were also upregulated by −P under each light treatment (Figure 5). Shading decreased the levels of *HK1*, *IDH*, and *SDH4* but increased *GS1* and *P4H* under P-deficient treatment conditions (Appendix A). Expression levels of the selected secondary-metabolite-related genes were upregulated by −P in each light treatment group (Appendix A). Shading decreased the level of *FLS* but increased the level of *UFGT* under −P treatment conditions.

### 2.5. Heatmap Correlation Clustering of Metabolites and Genes with Various Light and P Levels

Figure 7 shows the correlation heatmap of biosynthetic pathway metabolites and genes under different light and P levels. The transcription factors *PHO1* and *SPX2* were clustered into the same hierarchical group, indicating similar correlation expression patterns under P sufficient conditions (Figure 8A). *SPX2* was clustered with the transporters *GSTb* and *SWEET3* under −P and shaded treatment conditions (Figure 8A,B). PPP, AAG, and PTT-related metabolites were negatively correlated with isoflavone-related gene expression under P sufficient conditions. *UGT78D1* and *UGT78D2* showed similar expression patterns under P sufficient conditions with the transporters *SWEET3* and *AAP* and were further clustered with *ANR*, *IDH*, and *PHR1*. Similarly, under P-deficient conditions, IPM, PPP, AAG, and PTT were positively correlated with the transcription factor *SPX2*, transporter genes *GSTb* and *SWEET3*, secondary metabolic genes *LDOX*, *ANR,* and *UFGT* and primary-metabolite-related genes *ADT1* and *GS1*. The transcription factors *PHR1*, *PHO1*, and *SPX2* were clustered into the same hierarchical group, indicating similar correlation expression patterns under all light conditions. However, *GSTb* showed a different analogous correlation pattern under shaded conditions. Under low light conditions, PPP-related genes were negatively correlated and clustered with all transcription-factor- and transporter-related genes. Some genes (*ADT1*, *RPIA*, and *F3′5′H*) shared similar correlation expression in one group, while under low light conditions, *UGT75L12* and *UGT78D1* showed similar analogous patterns to other groups.

### 2.6. Overview of Metabolic PATHWAY and Gene Relation

There were significant negative correlations of phosphorus with *GSTb*, *SWEET3,* and *SPX2* under −P treatment conditions. The biomass had significant negative correlations with *SPX2* and *SWEET3* under −P treatment conditions (Appendix A). Under full light conditions, phosphorus had significant negative correlations with *AAP*, PTT biosynthesis, *SWEET3*, *SPX2*, and IPM but positive relationships with PPP and biomass under all light conditions (Figure 8 and Appendix A). Phosphorus had strong, significantly negative correlations with AAB, *AAP*, *GSTb*, PTT, *SWEET3*, *SPX2*, and IPM but strong, significantly positive correlations with AAG and PPP under shaded conditions.

## 3. Discussion

Light availability greatly affects plant growth and organ development. Plants respond to light variation to regulate their photosynthetic rate and maintain a dynamic nutrient balance. Myo-inositol is incorporated into Inositol-Phosphate phosphatase 1 (EC 3.1.3.25, *IMPL1*) to form the substrate Myo-Inositol, which has roles in plant signal transduction and environmental responses in solute synthesis and is utilized for the synthesis of sphingolipid-signaling molecules via root cell elongation using salicylic acid and ethylene-dependent pathways [6]. In this study of Longjing43 tea plants, the increase in myo-inositol that occurred under medium light conditions accelerated the synthesis of its oxidation product and noncellulose compounds. Moreover, the expression of *IMPL1* induced by P deficiency increased the Myo-Inositol concentration under medium light conditions, suggesting that different signaling pathways act in response to P starvation under different light intensities. Inositol trisphosphate is a secondary messenger and is a fundamental component of plant signaling that is involved in the control of plant development, ABA signaling, and the direct link between the phosphoinositol pathway and light signaling [32]. Inositol trisphosphate acts on the inositol triphosphate receptor to release calcium into the cytoplasm [33]. The rapid increase in available Ca in the cytoplasm contributes to young shoot growth. PPP is a metabolic pathway that acts in parallel to glycolysis and generates NADPH and ribose 5-phosphate, which are precursors to nucleotide synthesis [34]. Additionally, d ribose 5-phosphate is phosphorylated from ribose and synthesized from d-ribulose 5-phosphate and plays essential roles in carbohydrate anabolism and catabolism [35]. Increased expression of *RPIA* and *RBKS* due to P deficiency resulted in the accumulation of d-ribose 5-phosphate (Figure 3 and Appendix A), whereas the d-ribose 5-phosphate concentration decreased, and the ribose concentration increased due to a P deficiency in ML. This induced *RBKS* expression, indicating a differential regulation pathway in ML compared with FL and LL. The increase in the metabolites d-ribulose 5-phosphate, d-gluconate 6-phosphate, and d-gluconate suggests that the pentose phosphate pathway runs parallel to glycolysis under ML conditions. d-glucuronic acid acts as a key molecule that binds to toxic substances via the pentose and glucuronate interconversions pathway. These results suggest that young shoots and leaves respond weakly and bind to toxic substances under P deficient conditions. The enzyme d-xylose reductase (EC 1.1.1.307) belongs to the aldose reductase family and catalyzes the reduction in d-xylose to the sugar alcohol xylitol in Chaetomium thermophilum [26]. Under P deficient conditions, there is an increase in l-xylulose, a compound that utilizes carbon as a source for ethanol production, promoting the formation of young shoots under shaded conditions. Arabitol is important for enabling sugar utilization in microorganisms [36]. l-arabinose is a polymer of cell wall saccharides in plants [37]. In this study, pentose and glucuronate interconversion increased under ML conditions, indicating that sugar consumption is high for cell wall polysaccharides and stress adaption [38]. d-fructose is transformed into mannose and then further synthesized by hexokinase to form mannose 6P [39]. d-fructose originates from carbon assimilated during photosynthesis and is used to synthesize d-mannose [17]. d-mannose is the major monosaccharide component of N-glycans and relies on having an ample supply of d-mannose 6-phosphate. Our results show that the d-fructose and d-mannose concentrations increased under ML conditions, whereas that of d-mannose 6-phosphate competitively decreased under LL conditions. This suggests that carbon partitioning assimilates d-fructose and partially relies on N-glycans under medium shading and limited P conditions.

Light intensity reduction is a common approach to improve the concentration of free amino acids in tea leaves. Glycine and serine are two interconvertible amino acids that play important roles in one-carbon metabolism. Our results support the view that carbon-metabolism increases the concentration of GSM metabolites in young shoots under light shading conditions. Ser is easily transported through the phloem and is essential for the synthesis of proteins, metabolism, and signaling molecules [40]. The increase in Ser in young shoots due to shading conditions indicates that it is synthesized in leaves through the photorespiratory pathway and could be supplied to nonphotosynthetic organs. The threonine concentration increased due to P starvation in the present study, which supports previous findings in rice [41]. A common change and increase in threonine under two shading conditions is protein synthesis repression. Gly accumulation was also noted in a study of P-deficient maize [42]. It was reported that the accumulation of free amino acids in dark-treated tea leaves is caused by the biosynthesis activation of amino acids [43]. In our study, the decrease in Thr under P deficient conditions was due to shading; however, Thr increased with full light intensity due to P starvation. Phe plays a pivotal role in plant growth and development [44], and the increase in Phe under ML conditions in our study agrees with previous findings in tea plants. Tyrosine is used as a substrate to synthesize numerous specialized metabolites, and Trp serves as a protein component and precursor to various secondary plant metabolites [45]. LIV forms a small group of branched-chain amino acids (BCAAs) due to their small branched hydrocarbon residues and aliphatic nature and is found in membrane-spanning protein domains [46]. Leu contains an α-amino group, an α-carboxylic acid group, and a side chain isobutyl group, making it a non-polar aliphatic amino acid that is able to make leucine-rich repeat proteins [47]. TCA cycle enzymes act as components of the electron transport chain where succinate can only be oxidized by *succinate dehydrogenase* (*SDH*) [48]. *SDH* activity depends on the light intensity, and *SDH4* expression decreased with increased shading under both P regimes in tea plants (Appendix A), which is in agreement with a previous study on arabidopsis [27]. The increased concentrations of oxalic acid, citrate, isocitrate, oxoglutarate, succinate, fumarate, and malate are in accordance with their flux regulation in photosynthetic tissues [49] under ML conditions. l-aspartate, in addition to constituting proteins and being an active residue in many enzymes [50], is a precursor to the biosynthesis of multiple biomolecules required for plant growth and defense, such as nucleotides, nicotinamide adenine dinucleotide (NAD), organic acids, and amino acids and their derived metabolites [51]. A strong correlation was found between Asp and P treatment in barley [52], which might result in protein degradation. In our study, Asp and P shared high principal component contributions in Longjing43 tea plants. In pear plants, l-glutamate occupies a central position in amino acid metabolism [53]. This acidic amino acid is formed by the activity of glutamate synthase, utilizing glutamine and 2-oxoglutarate in arabidopsis [54]. Glutamate metabolism into ornithine, arginine, proline, and polyamines is the major network of nitrogen-metabolizing pathways, and it also produces intermediates such as nitric oxide and γ-aminobutyric acid that play critical roles in tea plant development and stress [55]. *GS1*, localized in the cytosol, contributes to metabolic systems via the regulation of ammonium assimilation [56]. A decrease in the substrate l-glutamine and a reduction in *GS1* expression negatively affect amino acid homeostasis and plastid development in young shoots under shading and P sufficient conditions. It was found that proline declined under low P conditions [57], and shading treatment could also alter proline accumulation [16]. In our study, increased expression of the proline and phenylalanine metabolic genes *P4H* and *ADT1* corresponded with increased synthesis of the metabolites proline, hydroxyproline, and l-phenylalanine under conditions of medium light combined with P deficiency. These increases could contribute to tea astringency and aroma.

Flavonoids are widely distributed secondary metabolites with different metabolic functions in plants and are involved in plant growth [58], reproduction, and defense against abiotic and biotic stresses. In plants, the phenylpropanoid pathway serves as a rich source of metabolites for flavonoid production [59]. It is involved in plant defense and survival. The enrichment of the phenylpropanoid pathway due to light and the effect of P (Figure 1 and Appendix A) drives the production of *p*-Coumarylshikimic acid, *p*-coumaroylquinic acid, and caffeoyl-CoA in young shoots [60]. Our results support the proposal that, as inhibitors, 4-coumaroyl and caffeoyl shikimic acids facilitate the 3-hydroxylation of 4-coumaric acid and 4-coumaroyl shikimate acid [61].

The flavonol glycoside biosynthesis pathway is derived from the central flavonoid biosynthesis pathway from dihydroquercetin. The substrate quercetin is an abundant flavonoid with strong antioxidant [28], anti-inflammatory [62], and anti-proliferative activities [63]. Flavonoid 3-*O*-glycosylation is catalyzed by the *UGT78D* family, with *UGT78D1* using UDP-rhamnose and *UGT78D2* using UDP-glucose [64]. Under medium light conditions, the concentration of quercetin increased, while that of its substrate isoquercetin, which is synthesized by *UGT78D2*, decreased, indicating that UDP-glucose was highly consumed. Furthermore, in young shoots, the concentration of the substrate rutin, synthesized by *UGT78D1*, increased under exposure to low light conditions. 3-*O*-glycosylation led to the repression of flavanol biosynthesis at the initial stage [65] and induction at the final stage [64]. In our study, the anthocyanidin-biosynthesis-related genes *ANR* and *LDOX* responded differently to the response shown for *ANS* in previous research [66], where *ANS* gene expression and EGC were suppressed in the tea plant cultivar Dangui when exposed to dark treatment.

Untargeted flavonoid accumulation for adjustment is in agreement with previous findings under P starvation in rice [67]. Similarly, the decrease in EGC under shaded conditions corresponds to the results of a previous study [2,7]. *LAR*s promote catechin monomer biosynthesis and inhibit their polymerization [68]. Our results indicate that *LAR* inhibits the polymerization of (+)-gallocatechin under P deficient and shaded conditions. Additionally, delphinidin-3-glucoside, which is synthesized by *UFGT*, was downregulated under shaded conditions. *LDOX* is involved in anthocyanidin biosynthesis through the catalysis of the oxidation of leucoanthocyanidins to cyanidins in grape plants [69]. Furthermore, cyanidin 3-glucoside, which is involved in energy metabolism, is synthesized by *ANR* and increases in concentration under light-shading. Previous studies indicate that anthocyanin and its derivatives increase under P deficient conditions in wheat [70]. Carbohydrate decreases are revealed as stunted growth in banana plants [71]. In our study, we noticed that an increase in *ANR* was associated with a lowered carbohydrate abundance under low light conditions, and the anthocyanin content further decreased under P sufficient conditions. Furthermore, the strong linear relationship between *ANR* and P under medium light conditions enhanced the downstream metabolite flow.

In terms of targeted metabolites, under P deficient conditions, the Serine concentration decreased with increasing shade in young shoots due to nutritional adjustment and photosynthetic carbon assimilation. In ML, an increase in Serine in line with the P supply was detected in tea plants, which is contrary to what occurred in P-deficient maize leaves [72]. Generally, most amino acids were present in higher concentrations in shade-treated tea leaves than under full-light conditions [2]. The reduction in Phe by P starvation indicates an irreversible effect of PAL genes on L-phenylalanine ammonia, as its crucial roles in plant growth and defense against various types of stresses have been reported in potato [73]. In addition, several studies have shown that light promotes catechin synthesis [15] and shade treatment is effective for the inhibition of flavonoid biosynthesis in soybean [29]. Similarly, our results support the phenomena in tea leaves whereby decreased concentrations of catechins (ECG, EGC and EGCG) were detected by half-shading under P sufficient conditions.

The ability to sense nutrients modulates plant reprogramming and adaption to environmental stress [74]. For instance, the cellular Pi status activates post-translational modification by ubiquitination and maintains Pi homeostasis [75]. In rice, SPX2 senses inositol pyrophosphate and regulates Pi starvation responsive genes through a Pi sensing mechanism [76,77]. When cellular Pi is high, *OsSPX2* interacts with the regulator *OsPHR2*, which prevents *PHR2* from binding to the phosphate starvation-inducible gene motif [77]. In tea plants, most genes involved in metabolic pathways have been found to increase due to a phosphorus deficiency, while the transcription of *SPX2* is downregulated by phosphorus treatment, indicating that different networks of sensing and regulation operate between rice and tea plants. Environmental factors, including nutrients and light, could affect the expression of *AAP* (*amino acid permeases*) belonging to amino acid transporters which function in the uptake, long-distance transport, and partitioning of amino acids [78,79]. Our study revealed that an increased glutamate concentration in young shoots is positively correlated with *AAP* expression, and this was found to be regulated by P deficiency combined with shading. In addition, the concentration of l-glutamate was higher in young shoots than in mature leaves, indicating the transportation of Glu from source to sink. It has been reported *GSTb* has a weak affinity for epicatechin which was reduced by P deficiency, but the gallate concentration increases in young shoots [4,7]. Our results show that the epicatechin gallate increased in young shoots due to P deficiency, in agreement with previous findings. 

## 4. Materials and Methods

### 4.1. Plant Materials

Longjing43, an important green tea cultivar with high economic value, was planted in pots in the open field at the Tea Research Institute, Hangzhou. An artificial soil growth medium was prepared from thoroughly mixed pertile, vermiculite, and turf by volume in the ratio of 1:2:3 (1:2:3 v/v). Coarse sand (1/4) was mixed thoroughly in growth medium (1:1 w/w) to make the growth medium stable. Plants were exposed to full natural light (100%; FL), medium light (50% of full light; ML), and low light (20% of full light; LL) transmission. The light intensity, temperature, and humidity were measured weekly using a LUX meter (Appendix A). Two different types of black perforated nylon agronet were used to restrict the light intensity. Furthermore, P-sufficient (+P) treatment was set up by adding 0.1 mM of P as KH_2_PO_4_, and P-deficient (−P) treatment excluded P. There were four pot replicates for each treatment group and four plants in each pot. To ensure experimental consistency, uniform plants were used, and sampling was performed after one year of stable growth. Young shoots (a bud with the first leaf) and leaves (fifth leaf) were harvested six times (three times for each in two independent experiments) at fortnightly intervals from mid-March to June (Appendix A). Shoot samples were collected within a few minutes on the same day to avoid differences caused by temperature, water, and light. Plant tissues collected for analysis were divided into two: one part was put in liquid nitrogen and stored at −80 °C, whereas the other was dried for elemental analysis. After collecting the upper plant parts, the fresh fibrous root was pulled out, washed carefully with deionised water, dissected, and then stored at −80 °C for further analysis.

### 4.2. Processing of the Samples and Instrument Conditions for Untargeted Metabolomics Analysis

Frozen young shoots, leaves, and roots were ground into fine powder using liquid nitrogen. One hundred grams of each sample was finely milled and weighed to exactly 0.1 g in a 2 mL Eppendorf tube under chilled conditions using ice cubes and liquid nitrogen. For the metabolomic analysis based on UPLC-Q-TOF/MS, plant samples were extracted with 2 mL of a solvent mixture of 75% methanol and 1% formic acid for 10 min in an ultrasonic bath and then centrifuged for 15 min at 12,000 rpm and −4 °C with few modifications, as prescribed [7]. Two milliliter extracted samples were filtered through an 0.22 mm PTFE filter and injected into a 1 dram glass vessel. GC×GC-TOF/MS samples were extracted using 1000 mL of methanol-chloroform (3:1, v/v) solvent (Sigma-Aldrich Co., St. Louis, MO, USA) from the plant sample. Volumes of 10 mL of L-2-chlorophenylalanine (0.2 mg/mL in water) were vortexed (30 s) and then centrifuged at 12,000 rpm for 10 min at −4 °C, and the supernatant (400 mL) was dried with vacuum centrifugation without heating. The samples were frozen in liquid nitrogen, and 80 mL of methoxyamine was added (8 mg/mL in pyridine) to each dried sample, followed by vortexing for 1 min. Methoxymation and trimethylsilylation were performed as described previously [7], and the glass vials were sealed for sample analysis.

An Agilent GC 6890N gas chromatograph with a high-speed TOF mass spectrometer detector (Pegasus HT, Leco Co., St. Joseph, CA, USA) was used for sample analysis. A first dimension DB-5 MS capillary column (30 m × 250 μm i.d., 0.25 μm film thickness; J&W Scientific, Folsom, CA, USA) and a second-dimension column DB-17H 2.5 m × 0.1 mm I.D. of 0.1 μm film thickness were used. Helium was used as the carrier gas at a constant flow rate of 1 mL min^−1^. The GC oven temperature was initially set at 80 °C for 2 min and then raised to 180 °C for 10 min, 240 °C for 5 min, 290 °C for 5 min, and finally, held at 290 °C for 9 min. The secondary oven temperature was kept at 5 °C offset above the primary oven. Each 1 μL aliquot of the derived sample was injected in splitless mode with the injector temperature set at 270 °C. The transfer line and ion source temperatures were set at 260 °C and 200 °C, respectively. The mass spectrometry data were acquired with electron impact ionization (70 eV) at full scan mode (m/z 30–600). The dwell time for each scan was set at a rate of 20 spectra per second, and the solvent delay was 3 min.

Metabolites were obtained with an ultra-Performance liquid chromatograph (ACQUITY UPLC, Waters Corp., Milford, MA, USA), which was equipped with an Acquity HSS T 3 column (1.8 mm, 100 mm 62.1 mm; Milford, MA, USA) kept at 40 °C and connected to a quadruple-time flight mass spectrometer (Xevo G2-XS QTOF, Waters Corp.). The mobile phase A was 0.1% aqueous formic acid, and mobile phase B was acetonitrile containing 0.1% formic acid. Gradient elution with a flow rate of 0.4 mL/min was performed as follows: the gradient profile was 0 min, 4% B; from 0 to 1 min; 4–7% B at 1–3 min; 7–25% B at 3–5 min; 25–45% B at 5–8 min; 45–90% B at 8–10 min; 90–100% B at 10–12 min; and 100% B at 12–15 min. The sample injection volume was 2 μL, and the samples were kept at 6 °C during the analysis. After each injection, the needle was rinsed with 600 μL of weak wash solution (water/methanol, 90:10) and 200 μL of strong wash solution (methanol/water, 90:10). Mass spectra were acquired using electrospray ionization at positive and negative modes over the range of m/z 100−1700. The stability of the method was tested by performing 10 repeated injections of the mixed samples every 2 h. The maximal tolerated m/z deviation, minimum/maximum chromatographic peak width in consecutive scans and the allowable retention time deviations were set as 15 ppm, 5/20 s, and 2 s, respectively.

### 4.3. Quantifying Targeted Catechin and Amino Acids by HPLC

Samples were assigned and extracted as for UPLC-Q-TOF/MS for catechin determination [7]. Free amino acid samples were analyzed as per the instructions provided by the waters AccQ.Tag chemistry package. HPLC analysis was carried out using an e2695 connecting 2998 photodiode array detector system (Waters) injected with 10 µL and 25 µL sample solutions for catechin and free amino acids, respectively. For catechins, distilled water with 2% formic acid was used as mobile phase A. Mobile phase B consisted of the HPLC solvent ACN. The samples were eluted at a column temperature of 40 ± 1 °C and a flow rate of 1 mL/min and monitored at 278 nm. Similarly, for amino acids, AccQ.Tag eluent from waters was used as mobile phase A. Mobile phase B was can, and the column temperature was set at 37 ± 2 °C. The remainder of the procedure followed the method prescribed in the AccQ. Tag chemistry package instruction manual. Peaks of catechins were identified by comparing sample retention time to those of authentic standards, and amino acid peaks were as prescribed in the manual.

### 4.4. qRT-PCR and ICP AES Analysis

Extracted RNA was synthesized to cDNA using the PrimeScript Reagent Kit, and genes were quantified as cp values using a LightCycler 480 II real-time machine with LightCycler 480 Software version 4.0 (Roche, Mannheim, Germany). To integrate the corresponding changes, four different Pi responsive genes and 19 metabolic genes from different pathways were selected to formulate their impact response on the pathway and the overview enrichment ratio. The gene primer sequences used for qRT-PCR are shown in Appendix A and the reference genes *GAPDH* and *Actin* with a sample (FL + P) were used for normalization. The P concentration in freeze-dried plant samples was measured by ICP-AES following digestion with concentrated HNO_3_ and HClO_4_.

### 4.5. Data Processing, Analysis, and Visualization

The data are responses from two independent experiments with three biological replicates each. The entire metabolomics dataset was screened, and identification was performed prior to normalizing. Data files from GC×GC-TOF/MS were processed in Leco software (Leco Co., St. Joseph, MI, USA) and were deconvoluted using an Automatic Mass Spectral Deconvolution and Identification System (AMDIS). Corresponding chromatogram peaks were compared to the National Institute of Standards and Technology (NIST, FairCom Co., Gaithersburg, MD, USA), and peaks were annotated from accurate mass measurements using online metabolite databases. The metabolites were identified based on the actual mass, retention time, and isotopic distribution, and accurate mass measurements were confirmed from the Metlin online web-based database and our published literature [4].

Raw chromatographic data acquired from the UPLC-Q-TOF/MS analysis were processed by TransOmics (Waters Corp., Milford, MA, USA), and peaks were annotated from accurate mass measurements using online metabolite databases. Peaks were identified based on the actual mass, retention time, and isotopic distribution. After normalization (relative expression`n to FL + P), PCA analysis was performed using R package “metabo” on each plant organ under different light and P treatment conditions to determine any group separation (Appendix A). Similarly, HPLC and ICP-AES data were used to measure the contents of targeted metabolites and plant nutrients in different organs under different light and P treatment conditions. Furthermore, all spectrometric measurements were subjected to a two-way ANOVA to compare means of the single light effect, P effect, and their interactions using Tukey’s HSD with the R package “agricolae”. The enriched pathways, pathway impact, pathway map, bar plot, box plot, and circular plot were drawn using LaTeX “tkiz” package. Pearson’s correlations were performed using the R statistical package “pcorr”. The linear regression graph, bi-plot PCA figure, and heatmap figure were drawn collectively using the R and LaTeX programs.

## 5. Conclusions

Collectively, the P concentration of young shoots declined as the light intensity decreased from FL to LL, and this decrease was further aggravated by P deficiency. Meanwhile, the abundance of most free amino acids, which are important contributors to the delicate flavor of green tea, increased in young shoots following exposure to moderate shading and a low P supply. In contrast, the concentrations of flavonoids, such as EGCG, C, EC and EGC, which give tea its astringent flavor, decreased in young shoots under moderate shading conditions and were modulated by P in a complicated way. This comprehensive analysis of the widely used green tea cultivar Longjing43, following exposure to changing light intensities coupled with P depletion, provides baseline information for the improvement of tea quality and maintenance of the yield in the management of tea plantations.

## Figures and Tables

**Figure 1 ijms-23-15194-f001:**
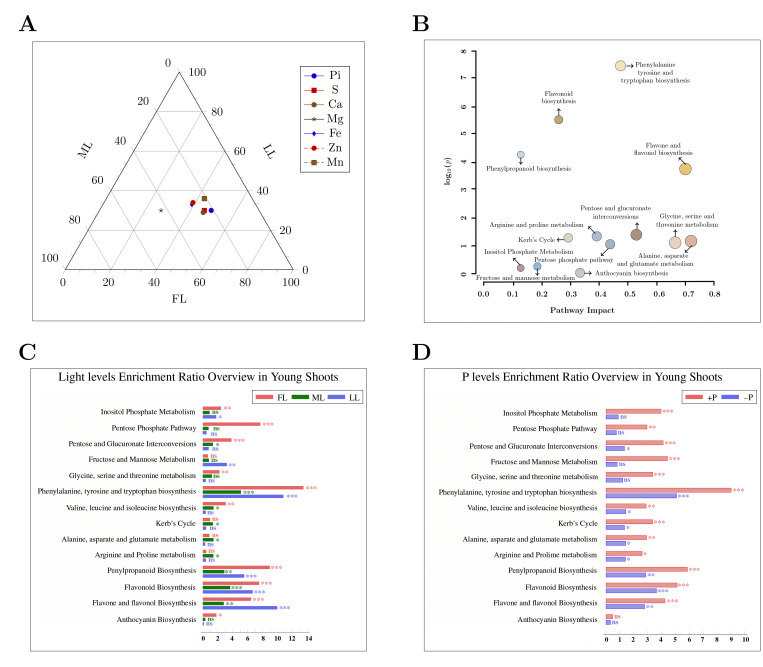
Overview of enriched pathways in young shoots of the Longjing43 cultivar. Ternary graph of plant nutrition (**A**) and pathway impact (**B**) under exposure to different light conditions with both P regimes. Overview of enriched pathways due to light conditions (**C**) and P levels (**D**) in young shoots. The circle with a pointed arrow shows the pathway impact under various light and P levels. Enrichment ratio bar graph asterisks indicate significant differences: *** = 0.001, ** = 0.01, * = 0.05, ns = non-significant differences. FL, full light intensity; ML, 50% of full light intensity; LL, 20% of full light intensity; +P, phosphorus-sufficient treatment; −P, phosphorus-deficient treatment.

**Figure 2 ijms-23-15194-f002:**
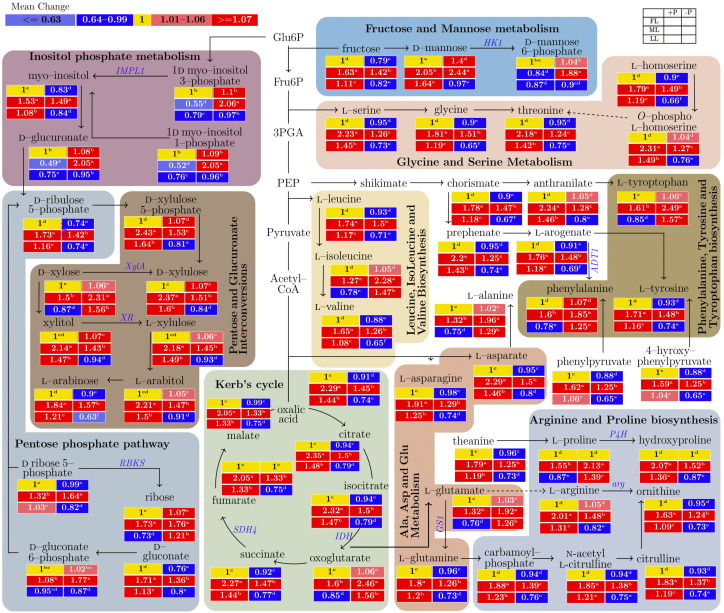
Interactive effects of light and P on the primary metabolites in young shoots of the Longjing43 cultivar measured by two-dimensional Gas Chromatography coupled with Time-of-Flight Mass Spectrometry (GC×GC-TOF/MS). Each metabolite mean followed by a different letter in the heatmap table indicates a significant difference. In the heatmap, the red boxes represent values of >1 and are separated into dark red and light red by the mean values of the figured pathway data (>1). Values of <1 are presented in blue, and mean values are separated into light blue and dark blue based on the figured pathway data (<1). The yellow boxes represent the normalized values. The blank table in the legend shows the positions of the mean P and light treatment values for the metabolites. FL, full light intensity; ML, 50% of full light intensity; LL, 20% of full light intensity; +P, phosphorus-sufficient treatment; −P, phosphorus-deficient treatment.

**Figure 3 ijms-23-15194-f003:**
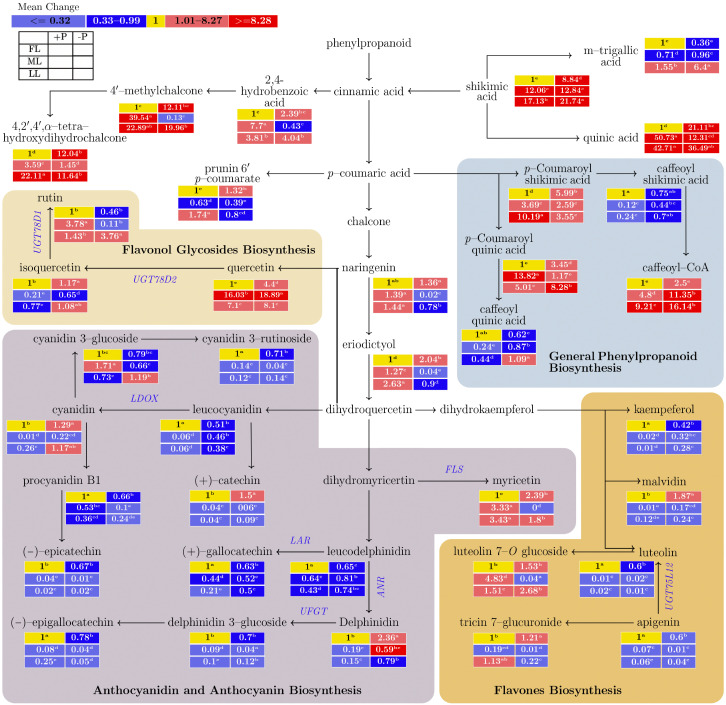
The effect of the interaction of light and P on the biosynthesis of secondary metabolites in young shoots of the Longjing43 cultivar measured by Ultra-Performance Liquid Chromatography-Quadrupole-Time of Flight Mass Spectrometry (UPLC-Q-TOF/MS). The mean of each metabolite followed by a different letter in the heatmap table indicates a significant difference due to light and P treatments. In the heatmap, red boxes represent values of >1 and are separated into dark and red and light red by the mean values of the figured pathway data (>1). The yellow boxes represent the normalized values. The blank table in the legend shows the positions of the mean P and light treatment values of the metabolites. FL, full light intensity; ML, 50% of full light intensity; LL, 20% of full light intensity; +P, phosphorus-sufficient treatment; −P, phosphorus-deficient treatment.

**Figure 4 ijms-23-15194-f004:**
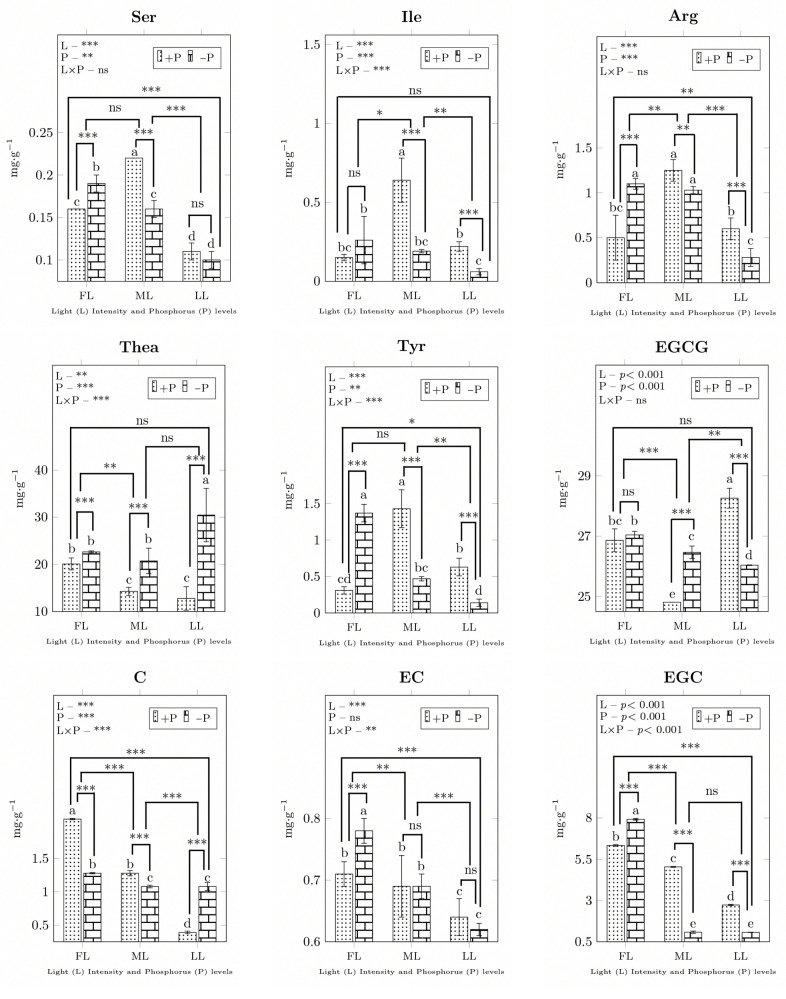
The mean concentrations of amino acids (Ser, Ile, Arg, Thea and Tyr) and flavonoids (EGCG, C, EC and EGC) in young shoots of the Longjing43 cultivar. Different letters above columns indicating significant difference. *** = 0.001, ** = 0.01, * = 0.05, ns = non-significant differences, between light and P interaction. FL, full light intensity; ML, 50% of full light intensity; LL, 20% of full light intensity; +P, phosphorus-sufficient treatment; −P, phosphorus-deficient treatment.

**Figure 5 ijms-23-15194-f005:**
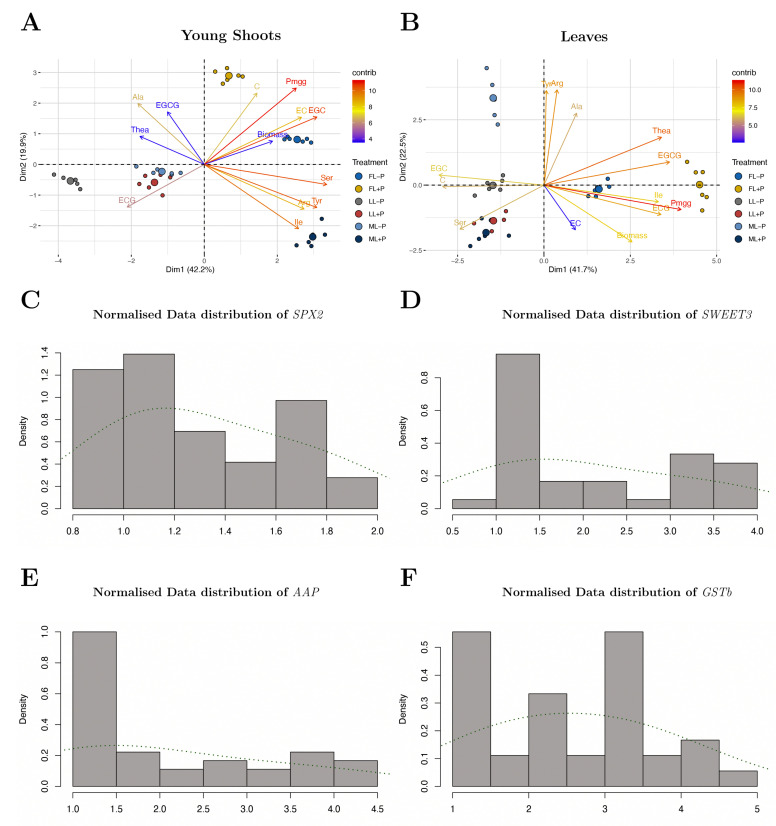
A single biplot for the dataset combines both samples and treatments of (**A**) young shoots and (**B**) leaves with the principal components. Histogram of the normalized data distribution under different light and P regimes for (**C**) *SPX2*, (**D**)*SWEET3*, (**E**) *AAP,* and (**F**) *GSTb* in young shoots. FL, full light intensity; ML, 50% of full light intensity; LL, 20% of full light intensity; +P, phosphorus-sufficient treatment; −P, phosphorus-deficient treatment.

**Figure 6 ijms-23-15194-f006:**
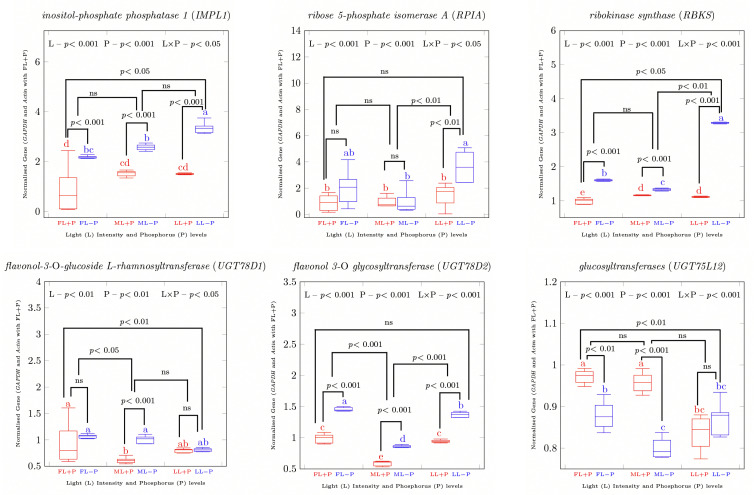
qRT-PCR was used to analyze the normalized primary and secondary metabolic gene transcripts (GAPDH and Actin) for the full light sample. Different letters above columns indicating significant difference and ns = non-significant differences between the light and P interaction. FL, full light intensity; ML, 50% of full light intensity; LL, 20% of full light intensity; +P, phosphorus-sufficient treatment; −P, phosphorus-deficient treatment.

**Figure 7 ijms-23-15194-f007:**
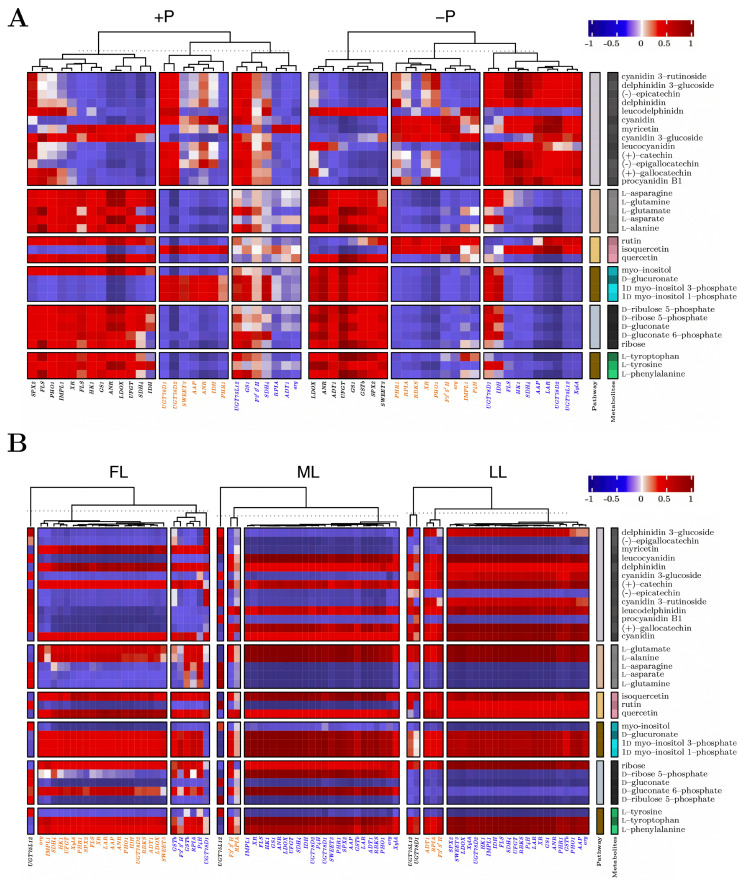
Heatmap correlation between pathway metabolites and genes under different (**A**) P levels and (**B**) light regimes. FL, full light intensity; ML, 50% of full light intensity; LL, 20% of full light intensity; +P, phosphorus-sufficient treatment; −P, phosphorus-deficient treatment.

**Figure 8 ijms-23-15194-f008:**
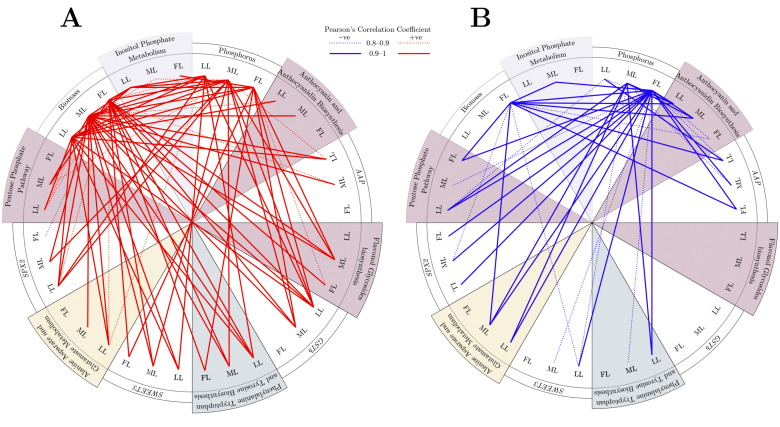
Circular representation of (**A**) positive and (**B**) negative correlations between pathways with phosphorus and biomass under different light regimes. FL, full light intensity; ML, 50% of full light intensity; LL, 20% of full light intensity; +P, phosphorus-sufficient treatment; −P, phosphorus-deficient treatment.

**Table 1 ijms-23-15194-t001:** Biomasses and concentrations of phosphorus in plants organs from the Longjing43 cultivar supplied with different light regimes and P levels.

Parameter	Organ	P Level	Light Intensity			Significance		
			FL	ML	LL	Light	P	L × P
Biomass	Young Shoots	+P	13.14 ± 0.65 a	12.42 ± 0.7 a	10.97 ± 0.94 b	*p* < 0.001	*p* < 0.001	*p* < 0.01
(g · plant^−1^)		−P	9.46 ± 0.34 c	8.44 ± 0.96 c	8.53 ± 0.73 c			
	Leaves	+P	15.16 ± 1.17 a	13.28 ± 1.47 b	12.99 ± 0.82 b	*p* < 0.001	*p* < 0.001	*p* < 0.01
		−P	12.47 ± 0.61 b	9.11 ± 0.75 c	8.29 ± 0.41 c			
	Root	+P	21.23 ± 1.35 bc	23.16 ± 0.78 a	22.06 ± 0.58 ab	*p* < 0.001	*p* < 0.001	*p* < 0.05
		−P	17.11 ± 0.92 d	20.29 ± 1.09 c	16.67 ± 0.52 d			
	Total	+P	49.53 ± 1.51 a	48.85 ± 2.45 a	46.01 ± 1.15 b	*p* < 0.001	*p* < 0.001	ns
		−P	39.03 ± 0.68 c	37.83 ± 1.69 c	33.49 ± 1.34 d			
P (mg · g^−1^)	Young Shoots	+P	10.97 ± 0.32 a	6.78 ± 0.31 c	5.21 ± 0.3 d	*p* < 0.001	*p* < 0.001	*p* < 0.01
		−P	8.27 ± 0.18 b	4.8 ± 0.26 d	3.13 ± 0.35 e			
	Leaves	+P	9.08 ± 0.34 a	4.2 ± 0.22 d	5.1 ± 0.2 c	*p* < 0.001	*p* < 0.001	*p* < 0.001
		−P	6.87 ± 0.21 b	3.29 ± 0.12 e	3.44 ± 0.3 e			

Note: Means with different letters in rows for the same nutritional content are significantly different. ns = non-significant differences for the light and P interaction. FL, full light intensity; ML, 50% of full light intensity; LL, 20% of full light intensity; +P, phosphorus-sufficient treatment; −P, phosphorus-deficient treatment.

## Data Availability

Not applicable.

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
