# Peer review of "Effect of Interactions between Phosphorus and Light Intensity on Metabolite Compositions in Tea Cultivar Longjing43"

_ijms, 2022, doi:10.3390/ijms232315194_

Round 1

Reviewer 1 Report

The manuscript is very important in field

But need improved language editing

Please repeat write conclusion

Please make part for stastical analysis 

Defined all abbreviations of all figure under the legend 

Author Response

Comments and Suggestions for Authors

The manuscript is very important in field

Re: Thanks for your positive valuation.

But need improved language editing

Re: The grammar of full-text has been checked and revised.

Please repeat write conclusion

Re: The conclusion has been rewritten in the resubmitted version according to your comment.

Please make part for stastical analysis

Re: The description of stastical analysis has been highlighted in M&M section (line 638-640).

Defined all abbreviations of all figure under the legend

Re: The abbreviations of all figures have been defined under the legend accordingly and the list of abbreviations has been added.

Reviewer 2 Report

1.) The title is not the most appropriate. It seems to be a phrase from within the paper and is totally inappropriate for the title of a scientific paper. I propose the following formula: The influence of the interactions between phosphorus and light intensity on the composition of metabolites in Longjing tea variety. The formula I propose is not unique nor mandatory, but this would be the way the title of the paper should be formulated.

2.) Discussions should respect the results obtained in the experiences described in the present paper. It is important for the reader to know whether similar experiences have been made before and to make a comparison with the present results and the interpretation of any differences.

3.) Thus, if the results presented in 2.1 show the biomass situation and the concentration of different minerals (K, S, Ca, Al, Fe, B, Cu, Mn and Zn), no results of this kind are shown in the discussions. If there is no such research in the specialized literature, although it is unlikely, to specify this aspect.

4.) During discussions, the species for which various results were obtained in the paper cited must be mentioned. There is a bit of confusion from this point of view.

5.) L 453-454. State under what conditions the ECG decrease was obtained? Only the shadow is referred to. In which species, in which varieties were these results obtained? In general, the authors do not specify varieties and species, although the differences between them can be significant.

6.) 5. 4.1 Why was the Longjing43 variety chosen for experiments? It is not explicitly mentioned where the tea plants used in the experiments were grown: in the open field or in the greenhouse? Some averages/sums of climate data should be presented, especially if plants have not been protected. Some mentions related to the origin of Longjing43 variety, agronomic and morphological particularities are required.

Author Response

1.) The title is not the most appropriate. It seems to be a phrase from within the paper and is totally inappropriate for the title of a scientific paper. I propose the following formula: The influence of the interactions between phosphorus and light intensity on the composition of metabolites in Longjing tea variety. The formula I propose is not unique nor mandatory, but this would be the way the title of the paper should be formulated.

Re: Thanks for your comments. We have revised the title according to your suggestion.

2.) Discussions should respect the results obtained in the experiences described in the present paper. It is important for the reader to know whether similar experiences have been made before and to make a comparison with the present results and the interpretation of any differences.  3.) Thus, if the results presented in 2.1 show the biomass situation and the concentration of different minerals (K, S, Ca, Al, Fe, B, Cu, Mn and Zn), no results of this kind are shown in the discussions. If there is no such research in the specialized literature, although it is unlikely, to specify this aspect.

Re: The results of biomass and mineral concentration have been shown in 2.1 as they were the basic information of plant samples harvested in this experiment. In that case, readers could easily understand the effect of treatments on plant growth. The reason for that this part of results have not been discussed was because we would like to foucus on the effect of phosphorus and light intensity on the composition of metabolites as shown in the title.

4.) During discussions, the species for which various results were obtained in the paper cited must be mentioned. There is a bit of confusion from this point of view.

Re: We have tried to clarified the species mentioned in the discussion section accordingly in the resubmitted version.

5.) L 453-454. State under what conditions the ECG decrease was obtained? Only the shadow is referred to. In which species, in which varieties were these results obtained? In general, the authors do not specify varieties and species, although the differences between them can be significant.

Re: The sentence has been rephrased according to your comments. We added the information about cultivar of tea plants and the treated condition. Please see line 476-479.

6.) 5. 4.1 Why was the Longjing43 variety chosen for experiments? It is not explicitly mentioned where the tea plants used in the experiments were grown: in the open field or in the greenhouse? Some averages/sums of climate data should be presented, especially if plants have not been protected. Some mentions related to the origin of Longjing43 variety, agronomic and morphological particularities are required.

Re: Longjing43 is an important and widely cultivated tea variety which is well-known and suiting for green tea with high economic value. The tea plants used in the experiments were grown in the open field. The reason and growth condition have been described in the section of M&M according to your comments. The climate data was shown in supplementary figure 1.

Reviewer 3 Report

1. Why do the authors use the cultivar Longjing43 for this study? Does it an important cultivar in tea plants or it has higher economic value than other cultivars? Or some other reasons need to be described in the section of M&M.

2. How to determine the optimal intensity of light and the concentration of phosphorus used in this study? Do these factors have a preliminary test?

3. Full names of abbreviations should be provided when the first time appears in the text including the abstract, for example, HPLC, etc.

4. L10-11: Needs to be rewritten because this sentence is unclear and misleading, particularly, the nature and role of light are very different from phosphorus. Don't mix these two factors in one sentence.

5. L11: and widely used to improve green tea quality - and is widely used to improve green tea quality

6. L29: the theaceae family - the Theaceae family

7. L30: tea plant - the tea plant

8. L31: The shade-tolerant plants possesses - The shade-tolerant plants possess

9. L33: its most high quality components - its most high-quality components

10. L78: involved on - involved in

11. The authors need to check grammar errors throughout the text.

12. Figure 6: How many replicates were conducted for each treatment in this experiment?

13. L480: plant reprogram - plant reprogramming

14. L486: most genes invovled in metabolic pathways were found increased - most genes involved in metabolic pathways were found to increase

Author Response

  1. Why do the authors use the cultivar Longjing43 for this study? Does it an important cultivar in tea plants or it has higher economic value than other cultivars? Or some other reasons need to be described in the section of M&M.

Re: Longjing43 is an important and widely cultivated tea variety which is well-known and suiting for green tea with high economic value. The reason has been described in the section of M&M according to your comment.

  1. How to determine the optimal intensity of light and the concentration of phosphorus used in this study? Do these factors have a preliminary test?

Re: The light intensity used as FL and the concentration of P as sufficient condition are the optimal for this study. The test has been done prior to the pot experiment in field conditions.

  1. Full names of abbreviations should be provided when the first time appears in the text including the abstract, for example, HPLC, etc.

Re: The full names of abbreviations in abstract and list of abbreviation have been provided in the resubmitted version.

  1. L10-11: Needs to be rewritten because this sentence is unclear and misleading, particularly, the nature and role of light are very different from phosphorus. Don't mix these two factors in one sentence.

Re: The sentences line10-11 in abstract have been replaced by the text below in the resubmitted version.

 ‘Light intensity influences energy production by increasing photosynthetic carbon, while phosphorus plays an important role in complex nucleic acid structure for regulation of protein synthesis. These two factors contribute to gene expression, metabolism and plant growth regulation, especially shading is an effective agronomic practice and widely used to improve green tea quality.’

  1. L11: and widely used to improve green tea quality - and is widely used to improve green tea quality

Re: The sentence has been revised with meaningful sentence in the resubmitted version.

  1. L29: the theaceae family - the Theaceae family

Re: The spelling has been revised accordingly in the resubmitted version.

  1. L30: tea plant - the tea plant

Re: The grammar of this sentence has been checked and revised accordingly in the new version.

  1. L31: The shade-tolerant plants possesses - The shade-tolerant plants possess

Re: The spelling has been revised accordingly in the resubmitted version.

  1. L33: its most high quality components - its most high-quality components

Re: The sentence has been revised accordingly in the resubmitted version.

  1. L78: involved on - involved in

Re: The grammar of this sentence has been checked and revised accordingly in the new version.

  1. The authors need to check grammar errors throughout the text.

Re: The grammar of full-text has been checked and revised.

  1. Figure 6: How many replicates were conducted for each treatment in this experiment?

Re: Four replicates were conducted for each treatment in our experiment and this has been emphasized in the revised manuscript (line 582).

  1. L480: plant reprogram - plant reprogramming

Re: The word has been revised accordingly in the resubmitted version.

  1. L486: most genes invovled in metabolic pathways were found increased - most genes involved in metabolic pathways were found to increase

Re: The sentence has been revised accordingly in the resubmitted version.